# Transductive Information Maximization
# For Few-Shot Learning

**Malik Boudiaf**[*]
ÉTS Montreal

**Ziko Imtiaz Masud**
ÉTS Montreal

**Jérôme Rony**
ÉTS Montreal

**Jose Dolz**
ÉTS Montreal

**Pablo Piantanida**
CentraleSupélec-CNRS
Université Paris-Saclay

**Ismail Ben Ayed**
ÉTS Montreal

## Abstract

We introduce Transductive Infomation Maximization (TIM) for few-shot learning. Our method maximizes the mutual information between the query features and their label predictions for a given few-shot task, in conjunction with a supervision loss based on the support set. Furthermore, we propose a new alternating-direction solver for our mutual-information loss, which substantially speeds up transductive-inference convergence over gradient-based optimization, while yielding similar accuracy. TIM inference is modular: it can be used on top of any base-training feature extractor. Following standard transductive few-shot settings, our comprehensive experiments[2] demonstrate that TIM outperforms state-of-the-art methods significantly across various datasets and networks, while used on top of a fixed feature extractor trained with simple cross-entropy on the base classes, without resorting to complex meta-learning schemes. It consistently brings between $2\%$ and $5\%$ improvement in accuracy over the best performing method, not only on all the well-established few-shot benchmarks but also on more challenging scenarios, with domain shifts and larger numbers of classes.

## 1   Introduction

Deep learning models have achieved unprecedented success, approaching human-level performances when trained on large-scale labeled data. However, the generalization of such models might be seriously challenged when dealing with new (unseen) classes, with only a few labeled instances per class. Humans, however, can learn new tasks rapidly from a handful of instances, by leveraging context and *prior* knowledge. The few-shot learning (FSL) paradigm [29, 8, 45] attempts to bridge this gap, and has recently attracted substantial research interest, with a large body of very recent works, e.g., [14, 7, 37, 49, 28, 5, 34, 19, 40, 48, 10, 38, 9], among many others. In the few-shot setting, a model is first trained on labeled data with *base* classes. Then, model generalization is evaluated on few-shot *tasks*, composed of unlabeled samples from novel classes unseen during training (the *query* set), assuming only one or a few labeled samples (the *support* set) are given per novel class.

Most of the existing approaches within the FSL framework are based on the "learning to learn" paradigm or meta-learning [9, 38, 45, 40, 22], where the training set is viewed as a series of balanced tasks (or *episodes*), so as to simulate test-time scenario. Popular works include prototypical networks [38], which describes each class with an embedding prototype and maximizes the log-probability of query samples via episodic training; matching network [45], which represents query predictions as

---

[*]Corresponding author: malik.boudiaf.1@etsmtl.net
[2]Code publicly available at `https://github.com/mboudiaf/TIM`

linear combinations of support labels and employs episodic training along with memory architectures; MAML [9], a meta-learner, which trains a model to make it "easy" to fine-tune; and the LSTM meta-learner in [35], which suggests optimization as a model for few-shot learning. A large body of meta-learning works followed-up lately, to only cite a few [37, 33, 30, 40, 49].

## 1.1   Related work

**Transductive inference:** In a recent line of work, *transductive* inference has emerged as an appealing approach to tackling few-shot tasks [7, 14, 19, 28, 34, 32, 27, 51], showing performance improvements over *inductive* inference. In the transductive setting[3], the model classifies the unlabeled query examples of a single few-shot task at once, instead of one sample at a time as in inductive methods. These recent experimental observations in few-shot learning are consistent with established facts in classical transductive inference [44, 18, 6], which is well-known to outperform inductive methods on small training sets. While [32] used information of unlabeled query samples via batch normalization, the authors of [28] were the first to model explicitly transductive inference in few-shot learning. Inspired by popular label-propagation concepts [6], they built a meta-learning framework that learns to propagate labels from labeled to unlabeled instances via a graph. The meta-learning transductive method in [14] used attention mechanisms to propagate labels to unlabeled query samples. More closely related to our work, the recent transductive inference of Dhillion et al. [7] minimizes the entropy of the network softmax predictions at unlabeled query samples, reporting competitive few-shot performances, while using standard cross-entropy training on the base classes. The competitive performance of [7] is in line with several recent inductive baselines [5, 46, 41], which reported that standard cross-entropy training for the base classes matches or exceeds the performances of more sophisticated meta-learning procedures. Also, the performance of [7] is in line with established results in the context of semi-supervised learning, where entropy minimization is widely used [11, 31, 2]. It is worth noting that the inference runtimes of transductive methods are, typically, much higher than their inductive counterparts. For, instance, the authors of [7] fine-tune all the parameters of a deep network during inference, which is several orders of magnitude slower than inductive methods such as ProtoNet [38]. Also, based on matrix inversion, the transductive inference in [28] has a complexity that is cubic in the number of query samples.

**Info-max principle:** While the semi-supervised and few-shot learning works in [11, 7] build upon Barlow's principle of entropy minimization [1], our few-shot formulation is inspired by the general info-max principle enunciated by Linsker [25], which formally consists in maximizing the Mutual Information (MI) between the inputs and outputs of a system. In our case, the inputs are the query features and the outputs are their label predictions. The idea is also related to info-max in the context of clustering [21, 16, 17]. More generally, info-max principles, well-established in the field of communications, were recently used in several deep-learning problems, e.g., representation learning [13, 43], metric learning [3] or domain adaptation [24], among other problems.

## 1.2   Contributions

- We propose Transductive Information Maximization (TIM) for few-shot learning. Our method maximizes the MI between the query features and their label predictions for a few-shot task at inference, while minimizing the cross-entropy loss on the support set.

- We derive an alternating-direction solver for our loss, which substantially speeds up transductive inference over gradient-based optimization, while yielding competitive accuracy.

- Following standard transductive few-shot settings, our comprehensive evaluations show that TIM outperforms state-of-the-art methods substantially across various datasets and networks, while using a simple cross-entropy training on the base classes, without complex meta-learning schemes. It consistently brings between $2\%$ and $5\%$ of improvement in accuracy over the best performing method, not only on all the well-established few-shot benchmarks but also on more challenging, recently introduced scenarios, with domain shifts and larger numbers of ways. Interestingly, our MI loss includes a label-marginal regularizer, which has a significant effect: it brings substantial improvements in accuracy, while facilitating optimization, reducing transductive runtimes by orders of magnitude.

## 2 Transductive Information Maximization

### 2.1 Few-shot setting

Assume we are given a labeled training set, $\mathcal{X}_{\text{base}} := \{x_i, y_i\}_{i=1}^{N_{\text{base}}}$, where $x_i$ denotes raw features of sample $i$ and $y_i$ its associated one-hot encoded label. Such labeled set is often referred to as the *meta-training* or *base* dataset in the few-shot literature. Let $\mathcal{Y}_{\text{base}}$ denote the set of classes for this base dataset. The few-shot scenario assumes that we are given a *test* dataset: $\mathcal{X}_{\text{test}} := \{x_i, y_i\}_{i=1}^{N_{\text{test}}}$, with a completely new set of classes $\mathcal{Y}_{\text{test}}$ such that $\mathcal{Y}_{\text{base}} \cap \mathcal{Y}_{\text{test}} = \emptyset$, from which we create randomly sampled few-shot *tasks*, each with a few labeled examples. Specifically, each $K$-way $N_S$-shot task involves sampling $N_S$ labeled examples from each of $K$ different classes, also chosen at random. Let $\mathcal{S}$ denote the set of these labeled examples, referred to as the *support* set with size $|\mathcal{S}| = N_S \cdot K$. Furthermore, each task has a *query* set denoted by $\mathcal{Q}$ composed of $|\mathcal{Q}| = N_Q \cdot K$ unlabeled (unseen) examples from each of the $K$ classes. With models trained on the base set, few-shot techniques use the labeled support sets to adapt to the tasks at hand, and are evaluated based on their performances on the unlabeled query sets.

### 2.2 Proposed formulation

We begin by introducing some basic notation and definitions before presenting our overall Transductive Information Maximization (TIM) loss and the different optimization strategies for tackling it. For a given few-shot task, with a support set $\mathcal{S}$ and a query set $\mathcal{Q}$, let $X$ denote the random variable associated with the raw features within $\mathcal{S} \cup \mathcal{Q}$, and let $Y \in \mathcal{Y} = \{1, \ldots, K\}$ be the random variable associated with the labels. Let $f_\phi : \mathcal{X} \longrightarrow \mathcal{Z} \subset \mathbb{R}^d$ denote the encoder (*i.e.*, feature-extractor) function of a deep neural network, where $\phi$ denotes the trainable parameters, and $\mathcal{Z}$ stands for the set of embedded features. The encoder is first trained from the base training set $\mathcal{X}_{\text{base}}$ using the standard cross-entropy loss, without any meta training or specific sampling schemes. Then, for each specific few-shot task, we propose to minimize a mutual-information loss defined over the query samples.

Formally, we define a soft-classifier, parametrized by weight matrix $\mathbf{W} \in \mathbb{R}^{K \times d}$, whose posterior distribution over labels given features[4], $p_{ik} := \mathbb{P}(Y = k | X = x_i; \mathbf{W}, \phi)$, and marginal distribution over query labels, $\widehat{p}_k = \mathbb{P}(Y_\mathcal{Q} = k; \mathbf{W}, \phi)$, are given by:

$$p_{ik} \propto \exp\left(-\frac{\tau}{2}\|w_k - z_i\|^2\right), \quad \text{and} \quad \widehat{p}_k = \frac{1}{|\mathcal{Q}|}\sum_{i \in \mathcal{Q}} p_{ik} \tag{1}$$

where $\mathbf{W} := [w_1, \ldots, w_K]$ denotes classifier weights, $z_i = \frac{f_\phi(x_i)}{\|f_\phi(x_i)\|_2}$ the L2-normalized embedded features, and $\tau$ is a temperature parameter.

Now, for each single few-shot task, we introduce our empirical weighted mutual information between the query samples and their latent labels, which integrates two terms: The first is an empirical (Monte-Carlo) estimate of the conditional entropy of labels given the query raw features, denoted $\widehat{\mathcal{H}}(Y_\mathcal{Q}|X_\mathcal{Q})$, while the second is the empirical label-marginal entropy, $\widehat{\mathcal{H}}(Y_\mathcal{Q})$.:

$$\widehat{\mathcal{I}}_\alpha(X_\mathcal{Q}; Y_\mathcal{Q}) := \underbrace{-\sum_{k=1}^{K} \widehat{p}_k \log \widehat{p}_k}_{\widehat{\mathcal{H}}(Y_\mathcal{Q}):\ \text{marginal entropy}} + \alpha \underbrace{\frac{1}{|\mathcal{Q}|}\sum_{i \in \mathcal{Q}}\sum_{k=1}^{K} p_{ik} \log(p_{ik})}_{-\widehat{\mathcal{H}}(Y_\mathcal{Q}|X_\mathcal{Q}):\ \text{conditional entropy}}, \tag{2}$$

with $\alpha$ a non-negative hyper-parameter. Notice that setting $\alpha = 1$ recovers the standard mutual information. Setting $\alpha < 1$ allows us to down-weight the conditional entropy term, whose gradients may dominate the marginal entropy's gradients as the predictions move towards the vertices of the simplex. The role of both terms in (2) will be discussed after introducing our overall transductive inference loss in the following, by embedding supervision from the task's support set.

We embed supervision information from support set $\mathcal{S}$ by integrating a standard cross-entropy loss CE with the information measure in Eq. (2), which enables us to formulate our Transductive Information

Maximization (**TIM**) loss as follows:

$$\boxed{\min_{\mathbf{W}} \; \lambda \cdot \mathrm{CE} - \widehat{\mathcal{I}}_\alpha(X_\mathcal{Q}; Y_\mathcal{Q})} \quad \text{with} \quad \mathrm{CE} \coloneqq -\frac{1}{|\mathcal{S}|} \sum_{i \in \mathcal{S}} \sum_{k=1}^{K} y_{ik} \log(p_{ik}), \tag{3}$$

where $\{y_{ik}\}$ denotes the $k^{th}$ component of the one-hot encoded label $\boldsymbol{y}_i$ associated to the $i$-th support sample. Non-negative hyper-parameters $\alpha$ and $\lambda$ will be fixed to $\alpha = \lambda = 0.1$ in all our experiments. It is worth to discuss in more details the role (importance) of the mutual information terms in (3):

- Conditional entropy $\widehat{\mathcal{H}}(Y_\mathcal{Q}|X_\mathcal{Q})$ aims at minimizing the uncertainty of the posteriors at unlabeled query samples, thereby encouraging the model to output *confident* predictions[5]. This entropy loss is widely used in the context of semi-supervised learning (SSL) [11, 31, 2], as it models effectively the *cluster* assumption: The classifier's boundaries should not occur at dense regions of the unlabeled features [11]. Recently, [7] introduced this term for few-shot learning, showing that entropy fine-tuning on query samples achieves competitive performances. In fact, if we remove the marginal entropy $\widehat{\mathcal{H}}(Y_\mathcal{Q})$ in objective (3), our TIM objective reduces to the loss in [7]. The conditional entropy $\widehat{\mathcal{H}}(Y_\mathcal{Q}|X_\mathcal{Q})$ is of paramount importance but its optimization requires special care, as its optima may easily lead to degenerate (non-suitable) solutions on the simplex vertices, mapping all samples to a single class. Such care may consist in using small learning rates and fine-tuning the whole network (which itself often contains several layers of regularization) as done in [7], both of which significantly slow down transductive inference.

- The label-marginal entropy regularizer $\widehat{\mathcal{H}}(Y_\mathcal{Q})$ encourages the marginal distribution of labels to be uniform, thereby avoiding degenerate solutions obtained when solely minimizing conditional entropy. Hence, it is highly important as it removes the need for implicit regularization, as mentioned in the previous paragraph. In particular, high-accuracy results can be obtained even using higher learning rates and fine-tuning only a fraction of the network parameters (classifier weights $\mathbf{W}$ instead of the whole network), speeding up substantially transductive runtimes. As it will be observed from our experiments, this term brings substantial improvements in performances (e.g., up to $10\%$ increase in accuracy over entropy fine-tuning on the standard few-shot benchmarks), while facilitating optimization, thereby reducing transductive runtimes by orders of magnitude.

### 2.3 Optimization

At this stage, we consider that the feature extractor has already been trained on base classes (using standard cross-entropy). We now propose two methods for minimizing our objective (3) for each test task. The first one is based on standard Gradient Descent (GD). The second is a novel way of optimizing mutual information, and is inspired by the Alternating Direction Method of Multipliers (ADMM). For both methods:

- The pre-trained feature extractor $f_\phi$ is kept fixed. Only the weights $\mathbf{W}$ are optimized for each task. Such a choice is discussed in details in subsection 3.4. Overall, and interestingly, we found that fine-tuning only classifier weights $\mathbf{W}$, while fixing feature-extractor parameters $\phi$, yielded the best performances for our mutual-information loss.

- For each task, weights $\mathbf{W}$ are initialized as the class prototypes of the support set:

$$\boldsymbol{w}_k^{(0)} = \frac{\sum_{i \in \mathcal{S}} y_{ik} \boldsymbol{z}_i}{\sum_{i \in \mathcal{S}} y_{ik}}$$

**Gradient descent (TIM-GD):** A straightforward way to minimize our loss in Eq. (3) is to perform gradient descent over $\mathbf{W}$, which we update using all the samples of the few-shot task (both support and query) at once (i.e., no mini-batch sampling). This gradient approach yields our overall best results, while being one order of magnitude faster than the transductive entropy-based fine-tuning in [7]. As will be shown later in our experiments, the method in [7] needs to fine-tune the whole network (i.e., to update both $\phi$ and $\mathbf{W}$), which provides implicit regularization, avoiding the degenerate solutions of entropy minimization. However, TIM-GD (with $\mathbf{W}$-updates only) still remains two

orders of magnitude slower than inductive closed-form solutions [38]. In the following, we present a more efficient solver for our problem.

**Alternating direction method (TIM-ADM):** We derive an Alternating Direction Method (ADM) for minimizing our objective in (3). Such scheme yields substantial speedups in transductive learning (one order of magnitude), while maintaining the high levels of accuracy of TIM-GD. To do so, we introduce auxiliary variables representing latent assignments of query samples, and minimize a mixed-variable objective by alternating two sub-steps, one optimizing w.r.t classifier's weights $\mathbf{W}$, and the other w.r.t the auxiliary variables $\boldsymbol{q}$.

**Proposition 1.** *The objective in (3) can be approximately minimized via the following constrained formulation of the problem:*

$$\min_{\mathbf{W},\boldsymbol{q}} \underbrace{-\frac{\lambda}{|\mathcal{S}|}\sum_{i\in\mathcal{S}}\sum_{k=1}^{K}y_{ik}\log(p_{ik})}_{\text{CE}} + \underbrace{\sum_{k=1}^{K}\widehat{q}_k\log\widehat{q}_k}_{\sim\widehat{\mathcal{H}}(Y_{\mathcal{Q}})} - \underbrace{\frac{\alpha}{|\mathcal{Q}|}\sum_{i\in\mathcal{Q}}\sum_{k=1}^{K}q_{ik}\log(p_{ik})}_{\sim\widehat{\mathcal{H}}(Y_{\mathcal{Q}}|X_{\mathcal{Q}})} + \underbrace{\frac{1}{|\mathcal{Q}|}\sum_{i\in\mathcal{Q}}\sum_{k=1}^{K}q_{ik}\log\frac{q_{ik}}{p_{ik}}}_{Penalty \equiv \mathcal{D}_{\text{KL}}(\mathbf{q}\|\mathbf{p})}$$

$$s.t \quad \sum_{k=1}^{K}q_{ik}=1, \quad q_{ik}\geq 0, \quad i\in\mathcal{Q}, \quad k\in\{1,\dots,K\}, \tag{4}$$

*where $\boldsymbol{q}=[q_{ik}]\in\mathbb{R}^{|\mathcal{Q}|\times K}$ are auxiliary variables, $\boldsymbol{p}=[p_{ik}]\in\mathbb{R}^{|\mathcal{Q}|\times K}$ and $\widehat{q}_k=\frac{1}{|\mathcal{Q}|}\sum_{i\in\mathcal{Q}}q_{ik}$.*

*Proof.* It is straightforward to notice that, when equality constraints $q_{ik}=p_{ik}$ are satisfied, the last term in objective (4), which can be viewed as a soft penalty for enforcing those equality constraints, vanishes. Objectives (3) and (4) then become equivalent. □

Splitting the problem into sub-problems on $\mathbf{W}$ and $\mathbf{q}$ as in Eq. (4) is closely related to the general principle of ADMM (Alternating Direction Method of Multipliers) [4], except that the KL divergence is not a typical penalty for imposing the equality constraints[6]. The main idea is to **decompose the original problem into two easier sub-problems**, one over $\mathbf{W}$ and the other over $\boldsymbol{q}$, which can be alternately solved, each in closed-form. Interestingly, this KL penalty is important as it completely removes the need for dual iterations for the simplex constraints in (4), yielding closed-form solutions:

**Proposition 2.** *ADM formulation in Proposition 1 can be approximately solved by alternating the following closed-form updates w.r.t auxiliary variables $\boldsymbol{q}$ and classifier weights $\mathbf{W}$ (t is the iteration index):*

$$q_{ik}^{(t+1)} \propto \frac{\left(p_{ik}^{(t)}\right)^{1+\alpha}}{\left(\sum_{i\in\mathcal{Q}}\left(p_{ik}^{(t)}\right)^{1+\alpha}\right)^{1/2}} \tag{5}$$

$$\boldsymbol{w}_k^{(t+1)} \leftarrow \frac{\frac{\lambda}{1+\alpha}\sum_{i\in\mathcal{S}}\left(y_{ik}\,\boldsymbol{z}_i + p_{ik}^{(t)}(\boldsymbol{w}_k^{(t)}-\boldsymbol{z}_i)\right) + \frac{|\mathcal{S}|}{|\mathcal{Q}|}\sum_{i\in\mathcal{Q}}\left(q_{ik}^{(t+1)}\boldsymbol{z}_i + p_{ik}^{(t)}(\boldsymbol{w}_k^{(t)}-\boldsymbol{z}_i)\right)}{\frac{\lambda}{1+\alpha}\sum_{i\in\mathcal{S}}y_{ik} + \frac{|\mathcal{S}|}{|\mathcal{Q}|}\sum_{i\in\mathcal{Q}}q_{ik}^{(t+1)}} \tag{6}$$

*Proof.* A detailed proof is deferred to the supplementary material. Here, we summarize the main technical ingredients of the approximation. Keeping the auxiliary variables $\mathbf{q}$ fixed, we optimize a convex approximation of Eq. (4) w.r.t $\mathbf{W}$. With $\mathbf{W}$ fixed, the objective is strictly convex w.r.t the auxiliary variables $\mathbf{q}$ whose updates come from a closed-form solution of the KKT (Karush–Kuhn–Tucker) conditions. Interestingly, the negative entropy of auxiliary variables, which appears in the penalty term, handles implicitly the simplex constraints, which removes the need for dual iterations to solve the KKT conditions. □

# 3   Experiments

**Hyperparameters:** To keep our experiments as simple as possible, **our hyperparameters are kept fixed across all the experiments and methods (TIM-GD and TIM-ADM)**. The conditional entropy weight $\alpha$ and the cross-entropy weights $\lambda$ in Objective (3) are both set to $0.1$. The temperature parameter $\tau$ in the classifier is set to $15$. In our TIM-GD method, we use the ADAM optimizer with the recommended parameters [20], and run 1000 iterations for each task. For TIM-ADM, we run 150 iterations.

**Base-training procedure:** The feature extractors are trained following the same simple base-training procedure as in [51] and using standard networks (ResNet-18 and WRN28-10), for all the experiments. Specifically, they are trained using the standard cross-entropy loss on the base classes, with label smoothing. The label-smoothing parameter is set to $0.1$. We emphasize that base training does not involve any meta-learning or episodic training strategy. The models are trained for 90 epochs, with the learning rate initialized to $0.1$, and divided by 10 at epochs 45 and 66. Batch size is set to 256 for ResNet-18, and to 128 for WRN28-10. During training, all the images are resized to $84 \times 84$, and we used the same data augmentation procedure as in [51], which includes random cropping, color jitter and random horizontal flipping.

**Datasets:** We resort to 3 few-shot learning datasets to benchmark the proposed models. As standard few-shot benchmarks, we use the *mini*-**Imagenet** [45] dataset, with 100 classes split as in [35], the **Caltech-UCSD Birds 200** [47] (CUB) dataset, with 200 classes, split following [5], and finally the larger *tiered*-**Imagenet** dataset, with 608 classes split as in [36].

## 3.1   Comparison to state-of-the-art

We first evaluate our methods TIM-GD and TIM-ADM on the widely adopted *mini*-ImageNet, *tiered*-ImageNet and *CUB* benchmark datasets, in the most common 1-shot 5-way and 5-shot 5-way scenarios, with 15 query shots for each class. Results are reported in Table 1, and are averaged over 10,000 episodes, following [46]. We can observe that both TIM-GD and TIM-ADM yield state-of-the-art performances, consistently across all standard datasets, scenarios and backbones, improving over both transductive and inductive methods, by significant margins.

## 3.2   Impact of domain-shift

Chen et al. [5] recently showed that the performance of most meta-learning methods may drop drastically when a domain-shift exists between the base training data and test data. Surprisingly, the simplest discriminative baseline exhibited the best performance in this case. Therefore, we evaluate our methods in this challenging scenario. To this end, we simulate a domain shift by training the feature encoder on *mini*-Imagenet while evaluating the methods on *CUB*, similarly to the setting introduced in [5]. TIM-GD and TIM-ADM beat previous methods by significant margins in the domain-shift scenario, consistently with our results in the standard few-shot benchmarks, thereby demonstrating an increased potential of applicability to real-world situations.

## 3.3   Pushing the meta-testing stage

Most few-shot papers only evaluate their method in the usual 5-way scenario. Nevertheless, [5] showed that meta-learning methods could be beaten by their discriminative baseline when more ways were introduced in each task. Therefore, we also provide results of our method in the more challenging 10-way and 20-way scenarios on *mini*-ImageNet. These results, which are presented in Table 3, show that TIM-GD outperforms other methods by significant margins, in both settings.

## 3.4   Ablation study

**Influence of each term:** We now assess the impact of each term[7] in our loss in Eq. (3) on the final performance of our methods. The results are reported in Table 4. We observe that integrating the three terms in our loss consistently outperforms any other configuration. Interestingly, removing

Table 1: Comparison to the state-of-the-art methods on *mini*-ImageNet, *tiered*-Imagenet and CUB. The methods are sub-grouped into transductive and inductive methods, as well as by backbone architecture. Our results (gray-shaded) are averaged over 10,000 episodes. "-" signifies the result is unavailable.

| Method | Transd. | Backbone | *mini*-ImageNet | | *tiered*-ImageNet | | CUB | |
|---|---|---|---|---|---|---|---|---|
| | | | 1-shot | 5-shot | 1-shot | 5-shot | 1-shot | 5-shot |
| MAML [9] | | ResNet-18 | 49.6 | 65.7 | - | - | 68.4 | 83.5 |
| RelatNet [40] | | ResNet-18 | 52.5 | 69.8 | - | - | 68.6 | 84.0 |
| MatchNet [45] | | ResNet-18 | 52.9 | 68.9 | - | - | 73.5 | 84.5 |
| ProtoNet [38] | | ResNet-18 | 54.2 | 73.4 | - | - | 73.0 | 86.6 |
| MTL [39] | ✗ | ResNet-12 | 61.2 | 75.5 | - | - | - | - |
| vFSL [50] | | ResNet-12 | 61.2 | 77.7 | - | - | - | - |
| Neg-cosine [26] | | ResNet-18 | 62.3 | 80.9 | - | - | 72.7 | 89.4 |
| MetaOpt [22] | | ResNet-12 | 62.6 | 78.6 | 66.0 | 81.6 | - | - |
| SimpleShot [46] | | ResNet-18 | 62.9 | 80.0 | 68.9 | 84.6 | 68.9 | 84.0 |
| Distill [41] | | ResNet-12 | 64.8 | 82.1 | 71.5 | 86.0 | - | - |
| RelatNet + T [14] | | ResNet-12 | 52.4 | 65.4 | - | - | - | - |
| ProtoNet + T [14] | | ResNet-12 | 55.2 | 71.1 | - | - | - | - |
| MatchNet+T [14] | | ResNet-12 | 56.3 | 69.8 | - | - | - | - |
| TPN [28] | | ResNet-12 | 59.5 | 75.7 | - | - | - | - |
| TEAM [34] | ✓ | ResNet-18 | 60.1 | 75.9 | - | - | - | - |
| Ent-min [7] | | ResNet-12 | 62.4 | 74.5 | 68.4 | 83.4 | - | - |
| CAN+T [14] | | ResNet-12 | 67.2 | 80.6 | 73.2 | 84.9 | - | - |
| LaplacianShot [51] | | ResNet-18 | 72.1 | 82.3 | 79.0 | 86.4 | 81.0 | 88.7 |
| TIM-ADM | | ResNet-18 | 73.6 | **85.0** | **80.0** | **88.5** | 81.9 | 90.7 |
| TIM-GD | | ResNet-18 | **73.9** | **85.0** | 79.9 | **88.5** | **82.2** | **90.8** |
| LEO [37] | | WRN28-10 | 61.8 | 77.6 | 66.3 | 81.4 | - | - |
| SimpleShot [46] | | WRN28-10 | 63.5 | 80.3 | 69.8 | 85.3 | - | - |
| MatchNet [45] | ✗ | WRN28-10 | 64.0 | 76.3 | - | - | - | - |
| CC+rot+unlabeled [10] | | WRN28-10 | 64.0 | 80.7 | 70.5 | 85.0 | - | - |
| FEAT [49] | | WRN28-10 | 65.1 | 81.1 | 70.4 | 84.4 | - | - |
| AWGIM [12] | | WRN28-10 | 63.1 | 78.4 | 67.7 | 82.8 | - | - |
| Ent-min [7] | | WRN28-10 | 65.7 | 78.4 | 73.3 | 85.5 | - | - |
| SIB [15] | | WRN28-10 | 70.0 | 79.2 | - | - | - | - |
| BD-CSPN [27] | ✓ | WRN28-10 | 70.3 | 81.9 | 78.7 | 86.92 | - | - |
| LaplacianShot [51] | | WRN28-10 | 74.9 | 84.1 | 80.2 | 87.6 | - | - |
| TIM-ADM | | WRN28-10 | 77.5 | 87.2 | 82.0 | 89.7 | - | - |
| TIM-GD | | WRN28-10 | **77.8** | **87.4** | **82.1** | **89.8** | - | - |

Table 2: The results for the domain-shift setting *mini*-Imagenet → CUB. The results obtained by our models (gray-shaded) are averaged over 10,000 episodes.

| Methods | Backbone | *mini*-ImageNet → CUB |
|---|---|---|
| | | 5-shot |
| MatchNet [45] | ResNet-18 | 53.1 |
| MAML [9] | ResNet-18 | 51.3 |
| ProtoNet [38] | ResNet-18 | 62.0 |
| RelatNet [40] | ResNet-18 | 57.7 |
| SimpleShot [46] | ResNet-18 | 64.0 |
| GNN [42] | ResNet-10 | 66.9 |
| Neg-Cosine [26] | ResNet-18 | 67.0 |
| Baseline [5] | ResNet-18 | 65.6 |
| LaplacianShot [51] | ResNet-18 | 66.3 |
| TIM-ADM | ResNet-18 | 70.3 |
| TIM-GD | ResNet-18 | **71.0** |

Table 3: Results for increasing the number of classes on *mini*-ImageNet. The results obtained by our models (gray-shaded) are averaged over 10,000 episodes.

| Methods | Backbone | 10-way | | 20-way | |
|---|---|---|---|---|---|
| | | 1-shot | 5-shot | 1-shot | 5-shot |
| MatchNet [45] | ResNet-18 | - | 52.3 | - | 36.8 |
| ProtoNet [38] | ResNet-18 | - | 59.2 | - | 45.0 |
| RelatNet [40] | ResNet-18 | - | 53.9 | - | 39.2 |
| SimpleShot [46] | ResNet-18 | 45.1 | 68.1 | 32.4 | 55.4 |
| Baseline [5] | ResNet-18 | - | 55.0 | - | 42.0 |
| Baseline++ [5] | ResNet-18 | - | 63.4 | - | 50.9 |
| TIM-ADM | ResNet-18 | 56.0 | **72.9** | **39.5** | 58.8 |
| TIM-GD | ResNet-18 | **56.1** | 72.8 | 39.3 | **59.5** |

Table 4: Ablation study on the effect of each term in our loss in Eq. (3), when only the classifier weights are fine-tuned, i.e., updating only $\mathbf{W}$, and when the whole network is fine-tuned, i.e., updating $\{\phi, \mathbf{W}\}$. The results are reported for ResNet-18 as backbone. The same term indexing as in Eq. (3) is used here: $\widehat{\mathcal{H}}(Y_{\mathcal{Q}})$: Marginal entropy, $\widehat{\mathcal{H}}(Y_{\mathcal{Q}}|X_{\mathcal{Q}})$: Conditional entropy, CE: Cross-entropy.

| Method | Param. | Loss | *mini*-ImageNet | | *tiered*-ImageNet | | CUB | |
|---|---|---|---|---|---|---|---|---|
| | | | 1-shot | 5-shot | 1-shot | 5-shot | 1-shot | 5-shot |
| TIM-ADM | $\{\mathbf{W}\}$ | CE | 60.0 | 79.6 | 68.0 | 84.6 | 68.6 | 86.4 |
| | | $CE + \widehat{\mathcal{H}}(Y_{\mathcal{Q}}|X_{\mathcal{Q}})$ | 36.0 | 77.0 | 48.1 | 82.5 | 48.5 | 86.5 |
| | | $CE - \widehat{\mathcal{H}}(Y_{\mathcal{Q}})$ | 66.7 | 82.0 | 74.0 | 86.5 | 74.2 | 88.3 |
| | | $CE - \widehat{\mathcal{H}}(Y_{\mathcal{Q}}) + \widehat{\mathcal{H}}(Y_{\mathcal{Q}}|X_{\mathcal{Q}})$ | **73.6** | **85.0** | **80.0** | **88.5** | **81.9** | **90.7** |
| TIM-GD | $\{\mathbf{W}\}$ | CE | 60.7 | 79.4 | 68.4 | 84.3 | 69.6 | 86.3 |
| | | $CE + \widehat{\mathcal{H}}(Y_{\mathcal{Q}}|X_{\mathcal{Q}})$ | 35.3 | 79.2 | 45.9 | 80.6 | 46.1 | 85.9 |
| | | $CE - \widehat{\mathcal{H}}(Y_{\mathcal{Q}})$ | 66.1 | 81.3 | 73.4 | 86.0 | 73.9 | 88.0 |
| | | $CE - \widehat{\mathcal{H}}(Y_{\mathcal{Q}}) + \widehat{\mathcal{H}}(Y_{\mathcal{Q}}|X_{\mathcal{Q}})$ | **73.9** | **85.0** | **79.9** | **88.5** | **82.2** | **90.8** |
| TIM-GD | $\{\phi, \mathbf{W}\}$ | CE | 60.8 | 81.6 | 65.7 | 83.5 | 68.7 | 87.7 |
| | | $CE + \widehat{\mathcal{H}}(Y_{\mathcal{Q}}|X_{\mathcal{Q}})$ | 62.7 | 81.9 | 66.9 | 82.8 | 72.6 | 89.0 |
| | | $CE - \widehat{\mathcal{H}}(Y_{\mathcal{Q}})$ | 62.3 | 82.7 | 68.3 | 85.4 | 70.7 | 88.8 |
| | | $CE - \widehat{\mathcal{H}}(Y_{\mathcal{Q}}) + \widehat{\mathcal{H}}(Y_{\mathcal{Q}}|X_{\mathcal{Q}})$ | **67.2** | **84.7** | **73.0** | **86.8** | **76.7** | **90.5** |

the label-marginal entropy, $\widehat{\mathcal{H}}(Y_{\mathcal{Q}})$, reduces significantly the performances in both TIM-GD and TIM-ADM, particularly when only classifier weights $\mathbf{W}$ are updated and feature extractor $\phi$ is fixed. Such a behavior could be explained by the following fact: the conditional entropy term, $\widehat{\mathcal{H}}(Y_{\mathcal{Q}}|X_{\mathcal{Q}})$, may yield degenerate solutions (assigning all query samples to a single class) on numerous tasks, when used alone. This emphasizes the importance of the label-marginal entropy term $\widehat{\mathcal{H}}(Y_{\mathcal{Q}})$ in our loss (3), which acts as a powerful regularizer to prevent such trivial solutions.

**Fine-tuning the whole network vs only the classifier weights:** While our TIM-GD and TIM-ADM optimize w.r.t $\mathbf{W}$ and keep base-trained encoder $f_\phi$ fixed at inference, the authors of [7] fine-tuned the whole network $\{\mathbf{W}, \phi\}$ when performing their transductive entropy minimization. To assess both approaches, we add to Table 4 a variant of TIM-GD, in which we fine-tune the whole network $\{\mathbf{W}, \phi\}$, by using the same optimization procedure as in [7]. We found that, besides being much slower, fine-tuning the whole network for our objective in Eq. 3 degrades the performances, as also conveyed by the convergence plots in Figure 1. Interestingly, when fine-tuning the whole network $\{\mathbf{W}, \phi\}$, the absence of $\widehat{\mathcal{H}}(Y_{\mathcal{Q}})$ in the entropy-based loss $CE + \widehat{\mathcal{H}}(Y_{\mathcal{Q}}|X_{\mathcal{Q}})$ does not cause the same drastic drop in performance as observed earlier when optimizing with respect to $\mathbf{W}$ only. We hypothesize that the network's intrinsic regularization (such as batch normalizations) and the use of small learning rates, as prescribed by [7], help the optimization process, preventing the predictions from approaching the vertices of the simplex, where entropy's gradients diverge.

## 3.5 Inference run-times

Transductive methods are generally slower at inference than their inductive counterparts, with run-times that are, typically, several orders of magnitude larger. In Table 5, we measure the average

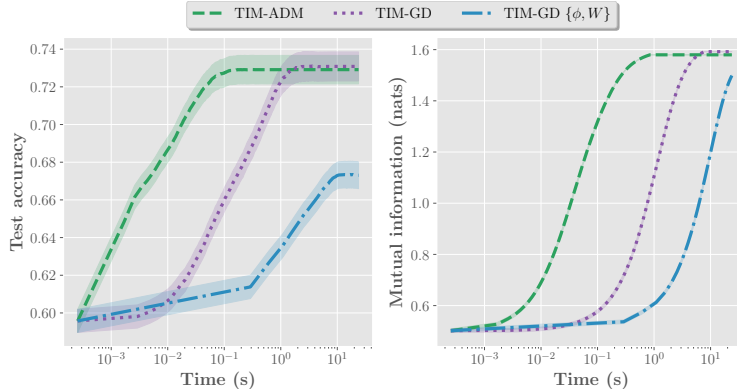

Figure 1: Convergence plots for our methods on *mini*-ImageNet with a ResNet-18. Solid lines are averages, while shadows are 95% confidence intervals. Time is in logarithmic scale. **Left:** Evolution of the test accuracy during transductive inference. **Right:** Evolution of the mutual information between query features and predictions $\widehat{\mathcal{I}}(X_{\mathcal{Q}}; Y_{\mathcal{Q}})$, computed as in Eq. (2), with $\alpha = 1$.

Table 5: Inference run-time per few-shot task for a 5-shot 5-way task on mini-ImageNet with a WRN28-10 backbone.

| | Run-times | | |
|---|---|---|---|
| Method | Parameters | Transductive | Inference/task (s) |
| SimpleShot [46] | $\{\mathbf{W}\}$ | ✗ | $9.0 \times 10^{-3}$ |
| TIM-ADM | $\{\mathbf{W}\}$ | ✓ | $1.2 \times 10^{-1}$ |
| TIM-GD | $\{\mathbf{W}\}$ | | $2.2 \times 10^{+0}$ |
| Ent-min [7] | $\{\phi, \mathbf{W}\}$ | | $2.1 \times 10^{+1}$ |

adaptation time per few-shot task, defined as the time required by each method to build the final classifier, for a 5-shot 5-way task on *mini*-ImageNet using the WRN28-10 network. Table 5 conveys that our ADM optimization gains one order of magnitude in run-time over our gradient-based method, and more than two orders of magnitude in comparison to [7], which fine-tunes the whole network. Note that TIM-ADM still remains slower than the inductive baseline. Our methods were run on the same GTX 1080 Ti GPU, while the run-time of [7] is directly reported from the paper.

## 4   Conclusion and future work

Our TIM inference establishes new state-of-the-art results on the standard few-shot benchmarks, as well as in more challenging scenarios, with larger numbers of classes and domain shifts. We used feature extractors based on a simple base-class training with the standard cross-entropy loss, without resorting to the complex meta-training schemes that are often used and advocated in the recent few-shot literature. TIM is modular: it could be plugged on top of any feature extractor and base training, regardless of how the training was conducted. Therefore, while we do not claim that the very challenging few-shot problem is solved, we believe that our model-agnostic TIM inference should be used as a strong baseline for future few-shot learning research. In future work, we target on giving a more theoretical ground for our proposed mutual-information objective, and on exploring further generalizations of the objective, e.g., via embedding domain-knowledge priors. Specifically, one of our theoretical goals will be to connect TIM's objective to the classifier's empirical risk on the query set, showing that the former could be viewed as a surrogate for the latter.

## 5   Acknowledgements

This research was supported by the National Science and Engineering Research Council of Canada (NSERC), via its Discovery Grant program.

**Broader impact**

Due to the simplicity and efficiency of our method, we lower the barrier of entry to few-shot learning. In turn, we think that it will make a wider breadth of real-world applications tractable. The impact (positive or negative) on society is similar to that of any other few-shot method: being only a tool, its impact is entirely dependent on the final applications, and on the intentions of the people and institutions deploying it.

In our strive towards finding simple and efficient formulations – for instance, we stick to a standard cross-entropy, which not only eases implementation, but also avoid the huge memory consumption of more complex methods – we believe our method can enable and empower persons and communities that are unable to afford the costly resources and infrastructures required. This may help level the playing field with larger and better funded entities. For instance, to be adapted to a new task, our TIM-ADM method requires a little more than a recent smartphone computational power. This could spawn a lot of fresh and new applications on edge devices, closer to the end-users, in real-time.

## Footnotes

[3]Transductive few-shot inference is not to be confused with semi-supervised few-shot learning [36, 23]. The latter uses extra unlabeled data during meta-training. Transductive inference has access to exactly the same training/testing data as its inductive counterpart.

[4]In order to simplify our notations, we deliberately omit the dependence of posteriors $p_{ik}$ on the network parameters $(\phi, \mathbf{W})$. Also, $p_{ik}$ takes the form of *softmax* predictions, but we omit the normalization constants.

[5]The global minima of each pointwise entropy in the sum of $\widehat{\mathcal{H}}(Y_\mathcal{Q}|X_\mathcal{Q})$ are one-hot vectors at the vertices of the simplex.

[6]Typically, ADMM methods use multiplier-based quadratic penalties for enforcing the equality constraint.

[7]The **W** and **q** updates of TIM-ADM associated to each configuration can be found in the supplementary material.

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
