[Supplementary Material]

# A    Proofs

**Proof of Proposition 1**

*Proof.* Let us start from the initial optimization problem:

$$\min_{\mathbf{W}} \quad \sum_{k=1}^{K} \widehat{p}_k \log \widehat{p}_k - \frac{\alpha}{|\mathcal{Q}|} \sum_{i \in \mathcal{Q}} \sum_{k=1}^{K} p_{ik} \log p_{ik} - \frac{\lambda}{|\mathcal{S}|} \sum_{i \in \mathcal{S}} \sum_{k=1}^{K} y_{ik} \log p_{ik} \tag{7}$$

We can reformulate problem (7) using the ADM approach, i.e., by introducing auxiliary variables $\boldsymbol{q} = [q_{ik}] \in \mathbb{R}^{|\mathcal{Q}| \times K}$ and enforcing equality constraint $\boldsymbol{q} = \boldsymbol{p}$, with $\boldsymbol{p} = [p_{ik}] \in \mathbb{R}^{|\mathcal{Q}| \times K}$, in addition to pointwise simplex constraints:

$$\min_{\mathbf{W}, \boldsymbol{q}} \quad \sum_{k=1}^{K} \widehat{q}_k \log \widehat{q}_k - \frac{\alpha}{|\mathcal{Q}|} \sum_{i \in \mathcal{Q}} \sum_{k=1}^{K} q_{ik} \log p_{ik} - \frac{\lambda}{|\mathcal{S}|} \sum_{i \in \mathcal{S}} \sum_{k=1}^{K} y_{ik} \log p_{ik}$$

$$\text{s.t.} \quad q_{ik} = p_{ik}, \quad i \in \mathcal{Q}, \quad k \in \{1, \ldots, K\}$$

$$\sum_{k=1}^{K} q_{ik} = 1, \quad i \in \mathcal{Q}$$

$$q_{ik} \geq 0, \quad i \in \mathcal{Q}, \quad k \in \{1, \ldots, K\} \tag{8}$$

We can slove constrained problem (8) with a penalty-based approach, which encourages auxiliary pointwise predictions $\boldsymbol{q}_i = [q_{i1}, \ldots, q_{iK}]$ to be close to our model's posteriors $\boldsymbol{p}_i = [p_{i1}, \ldots, p_{iK}]$. To add a penalty encouraging equality constraints $\boldsymbol{q}_i = \boldsymbol{p}_i$, we use the Kullback–Leibler (KL) divergence, which is given by:

$$\mathcal{D}_{\text{KL}}(\boldsymbol{q}_i \| \boldsymbol{p}_i) = \sum_{k=1}^{K} q_{ik} \log \frac{q_{ik}}{p_{ik}}$$

Thus, our constrained optimization problem becomes:

$$\min_{\mathbf{W}, \boldsymbol{q}} \quad \sum_{k=1}^{K} \widehat{q}_k \log \widehat{q}_k - \frac{\alpha}{|\mathcal{Q}|} \sum_{i \in \mathcal{Q}} \sum_{k=1}^{K} q_{ik} \log p_{ik} - \frac{\lambda}{|\mathcal{S}|} \sum_{i \in \mathcal{S}} \sum_{k=1}^{K} y_{ik} \log p_{ik} + \frac{1}{|\mathcal{Q}|} \sum_{i \in \mathcal{Q}} \mathcal{D}_{\text{KL}}(\boldsymbol{q}_i \| \boldsymbol{p}_i)$$

$$\text{s.t.} \quad \sum_{k=1}^{K} q_{ik} = 1, \quad i \in \mathcal{Q}$$

$$q_{ik} \geq 0, \quad i \in \mathcal{Q}, \quad k \in \{1, \ldots, K\} \tag{9}$$

$\square$

**Proof of Proposition 2**

*Proof.* Recall that we consider a softmax classifier over distances to weights $\mathbf{W} = \{\boldsymbol{w}_1, \ldots, \boldsymbol{w}_K\}$. To simplify the notations, we will omit the dependence upon $\phi$ in what follows, and write $\boldsymbol{z}_i = \frac{f_\phi(\boldsymbol{x}_i)}{\|f_\phi(\boldsymbol{x}_i)\|}$, such that:

$$p_{ik} = \frac{e^{-\frac{\tau}{2} \|\boldsymbol{z}_i - \boldsymbol{w}_k\|^2}}{\sum_{j=1}^{K} e^{-\frac{\tau}{2} \|\boldsymbol{z}_i - \boldsymbol{w}_j\|^2}} \tag{10}$$

Without loss of generality, we use $\tau = 1$ in what follows. Plugging the expression of $p_{ik}$ into Eq. (4), and grouping terms together, we get:

$$(4) = \sum_{k=1}^{K} \widehat{q}_k \log \widehat{q}_k - \frac{1+\alpha}{|\mathcal{Q}|} \sum_{i \in \mathcal{Q}} \sum_{k=1}^{K} q_{ik} \log p_{ik} - \frac{\lambda}{|\mathcal{S}|} \sum_{i \in \mathcal{S}} \sum_{k=1}^{K} y_{ik} \log p_{ik} + \frac{1}{|\mathcal{Q}|} \sum_{i \in \mathcal{Q}} \sum_{k=1}^{K} q_{ik} \log q_{ik}$$

$$= \sum_{k=1}^{K} \widehat{q}_k \log \widehat{q}_k$$

$$+ \frac{1+\alpha}{2|\mathcal{Q}|} \sum_{i \in \mathcal{Q}} \sum_{k=1}^{K} q_{ik} \|\boldsymbol{z}_i - \boldsymbol{w}_k\|^2 + \frac{1+\alpha}{|\mathcal{Q}|} \sum_{i \in \mathcal{Q}} \log \left( \sum_{j=1}^{K} e^{-\frac{1}{2}\|\boldsymbol{z}_i - \boldsymbol{w}_j\|^2} \right)$$

$$+ \frac{\lambda}{2|\mathcal{S}|} \sum_{i \in \mathcal{S}} \sum_{k=1}^{K} y_{ik} \|\boldsymbol{z}_i - \boldsymbol{w}_k\|^2 + \frac{\lambda}{|\mathcal{S}|} \sum_{i \in \mathcal{S}} \log \left( \sum_{j=1}^{K} e^{-\frac{1}{2}\|\boldsymbol{z}_i - \boldsymbol{w}_j\|^2} \right) \qquad (11)$$

$$+ \frac{1}{|\mathcal{Q}|} \sum_{i \in \mathcal{Q}} \sum_{k=1}^{K} q_{ik} \log q_{ik}$$

Now, we can solve our problem approximately by alternating two sub-steps: one sub-step optimizes w.r.t classifier weights $\mathbf{W}$ while auxiliary variables $\boldsymbol{q}$ are fixed; another sub-step fixes $\mathbf{W}$ and update $\boldsymbol{q}$.

- $\mathbf{W}$-update: Omitting the terms that do not involve $\mathbf{W}$, Eq. (11) reads:

$$\underbrace{\frac{\lambda}{2|\mathcal{S}|} \sum_{i \in \mathcal{S}} y_{ik} \|\boldsymbol{z}_i - \boldsymbol{w}_k\|^2 + \frac{1+\alpha}{2|\mathcal{Q}|} \sum_{i \in \mathcal{Q}} q_{ik} \|\boldsymbol{z}_i - \boldsymbol{w}_k\|^2}_{\mathcal{C}\text{:convex}}$$

$$+ \underbrace{\frac{\lambda}{|\mathcal{S}|} \sum_{i \in \mathcal{S}} \log \left( \sum_{j=1}^{K} e^{-\frac{1}{2}\|\boldsymbol{z}_i - \boldsymbol{w}_j\|^2} \right) + \frac{1+\alpha}{|\mathcal{Q}|} \sum_{i \in \mathcal{Q}} \log \left( \sum_{j=1}^{K} e^{-\frac{1}{2}\|\boldsymbol{z}_i - \boldsymbol{w}_j\|^2} \right)}_{\bar{\mathcal{C}}\text{:non-convex}} \qquad (12)$$

One can notice that objective (11) is not convex w.r.t $\boldsymbol{w}_k$. Actually, it can be split into convex and non-convex parts as in Eq. (12). Thus, we cannot simply set the gradients to 0 to get the optimal $\boldsymbol{w}_k$. The non-convex part can be linearized at current solution $\boldsymbol{w}_k^{(t)}$ as follows:

$$\bar{\mathcal{C}}(\boldsymbol{w}_k) \approx \bar{\mathcal{C}}(\boldsymbol{w}_k^{(t)}) + \frac{\partial \bar{\mathcal{C}}}{\partial \boldsymbol{w}_k}(\boldsymbol{w}_k^{(t)})^T (\boldsymbol{w}_k - \boldsymbol{w}_k^{(t)})$$

$$\overset{\mathrm{c}}{=} \frac{\lambda}{|\mathcal{S}|} \sum_{i \in \mathcal{S}} p_{ik}^{(t)} (\boldsymbol{z}_i - \boldsymbol{w}_k^{(t)})^T \boldsymbol{w}_k + \frac{1+\alpha}{|\mathcal{Q}|} \sum_{i \in \mathcal{Q}} p_{ik}^{(t)} (\boldsymbol{z}_i - \boldsymbol{w}_k^{(t)})^T \boldsymbol{w}_k \qquad (13)$$

Where $\overset{\mathrm{c}}{=}$ stands for "equal, up to an additive constant". By adding this linear term to the convex part $\mathcal{C}$, we can obtain a strictly convex objective in $\boldsymbol{w}_k$, whose gradients w.r.t $\boldsymbol{w}_k$ read:

$$\frac{\partial (12)}{\partial \boldsymbol{w}_k} \approx \frac{\lambda}{|\mathcal{S}|} [\sum_{i \in \mathcal{S}} y_{ik}(\boldsymbol{z}_i - \boldsymbol{w}_k) + p_{ik}^{(t)}(\boldsymbol{z}_i - \boldsymbol{w}_k^{(t)})] +$$

$$\frac{1+\alpha}{|\mathcal{Q}|} [\sum_{i \in \mathcal{Q}} q_{ik}(\boldsymbol{z}_i - \boldsymbol{w}_k) + p_{ik}^{(t)}(\boldsymbol{z}_i - \boldsymbol{w}_k^{(t)})] \qquad (14)$$

Note that the approximation we do here is similar in spirit to concave-convex procedures, which are well known in optimization. Concave-convex techniques proceed as follows: for a function in the form of a sum of a concave term and a convex term, the concave part is replaced by its first-order approximation, while the convex part is kept as is. The difference here is that the part that we linearize in Eq. (12) is not concave. Setting the gradients above to 0 yields the optimal solution for the approximate objective.

Another solution to obtain a strictly convex objective would have been to discard the non-convex part $\bar{\mathcal{C}}$. Very interestingly, in this case, one would recover $\boldsymbol{w}_k$ updates that would very much resemble the prototype updates of the K-means clustering algorithm (slightly modified to take into account the fact that for support points in $\mathcal{S}$ have labels). Note that the link between regularized K-means and mutual information maximization has been extensively explored in [17]. Of course, in this case, the approximation is not as good as the first-order approximation above, and we found that omitting the non-convex part might decrease the performances significantly.

- $q$-update: With weights $\mathbf{W}$ fixed, the objective is convex w.r.t auxiliary variables $\boldsymbol{q}_i$ (sum of linear and convex functions) and the simplex constraints are affine. Therefore, one can minimize this constrained convex problem for each $\boldsymbol{q}_i$ by solving the Karush-Kuhn-Tucker (KKT) conditions[8]. The KKT conditions yield closed-form solutions for both primal variable $\boldsymbol{q}_i$ and the dual variable (Lagrange multiplier) corresponding to simplex constraint $\sum_{j=1}^{K} q_{ij} = 1$. Interestingly, the negative entropy of auxiliary variables, i.e., $\sum_{k=1}^{K} q_{ik} \log q_{ik}$, which appears in the penalty term, handles implicitly non-negativity constraints $\boldsymbol{q}_i \geq 0$. In fact, this negative entropy acts as a barrier function, restricting the domain of each $\boldsymbol{q}_i$ to non-negative values, which avoids extra dual variables and Lagrangian-dual inner iterations for constraints $\boldsymbol{q}_i \geq 0$. As we will see, the closed-form solutions of the KKT conditions satisfy these non-negativity constraints, without explicitly imposing them. In addition to non-negativity, for each point $i$, we need to handle probability simplex constraints $\sum_{k=1}^{K} q_{ik} = 1$. Let $\gamma_i \in \mathbb{R}$ denote the Lagrangian multiplier corresponding to this constraint. The KKT conditions correspond to setting the following gradient of the Lagrangian function to zero, while enforcing the simplex constraints:

$$\frac{\partial(4)}{\partial q_{ik}} = -\frac{1+\alpha}{|\mathcal{Q}|} \log p_{ik} + \frac{1}{|\mathcal{Q}|} (\log \widehat{q}_k + 1) + \frac{1}{|\mathcal{Q}|} (\log q_{ik} + 1) + \gamma_i \tag{15}$$

$$= \frac{1}{|\mathcal{Q}|} \left( \log(\frac{q_{ik} \widehat{q}_k}{p_{ik}^{1+\alpha}}) + 2 \right) + \gamma_i \tag{16}$$

This yields:

$$q_{ik} = \frac{p_{ik}^{1+\alpha}}{\widehat{q}_k} e^{-(\gamma_i |\mathcal{Q}|+2)} \tag{17}$$

Applying simplex constraint $\sum_{j=1}^{K} q_{ij} = 1$ to (17), Lagrange multiplier $\gamma_i$ verifies:

$$e^{-(\gamma_i |\mathcal{Q}|+2)} = \frac{1}{\displaystyle\sum_{j=1}^{K} \frac{p_{ij}^{1+\alpha}}{\widehat{q}_j}} \tag{18}$$

Hence, plugging (18) in (17) yields:

$$q_{ik} = \frac{\dfrac{p_{ik}^{1+\alpha}}{\widehat{q}_k}}{\displaystyle\sum_{j=1}^{K} \frac{p_{ij}^{1+\alpha}}{\widehat{q}_j}} \tag{19}$$

Using the definition of $\widehat{q}_k$, we can decouple this equation:

$$\widehat{q}_k = \frac{1}{|\mathcal{Q}|} \sum_{i \in \mathcal{Q}} q_{ik} \propto \sum_{i \in \mathcal{Q}} \frac{p_{ik}^{1+\alpha}}{\widehat{q}_k} \tag{20}$$

which implies:

$$\widehat{q}_k \propto \left( \sum_{i \in \mathcal{Q}} p_{ik}^{1+\alpha} \right)^{1/2} \tag{21}$$

Plugging this back in Eq. (19), we get:

$$q_{ik} \propto \frac{p_{ik}^{1+\alpha}}{\left(\sum_{i \in \mathcal{Q}} p_{ik}^{1+\alpha}\right)^{1/2}} \tag{22}$$

Notice that $q_{ik} \geq 0$, hence the solution fulfils the positivity constraint of the original problem.

$\square$

# B   TIM algorithms

In this section, we provide the pseudo-code for TIM's inference stage (both TIM-GD and TIM-ADM).

---

**Algorithm 1:** TIM-ADM

**Input** : Pre-trained encoder $f_\phi$, Task $\{\mathcal{S}, \mathcal{Q}\}$, # iterations $iter$, Temperature $\tau$, Weights $\{\lambda, \alpha\}$

$\boldsymbol{z}_i \leftarrow \frac{f_\phi(\boldsymbol{x}_i)}{\|f_\phi(\boldsymbol{x}_i)\|_2} , i \in \mathcal{S} \cup \mathcal{Q}$

$\boldsymbol{w}_k \leftarrow \frac{\sum_{i \in \mathcal{S}} y_{ik} \boldsymbol{z}_i}{\sum_{i \in \mathcal{S}} y_{ik}} , k \in \{1, \ldots, K\}$

**for** $i \leftarrow 0$ **to** $iter$ **do**

    $p_{ik} \leftarrow \exp\left(-\frac{\tau}{2} \|\boldsymbol{w}_k - \boldsymbol{z}_i\|^2\right), i \in \mathcal{S} \cup \mathcal{Q}$

    $p_{ik} \leftarrow \frac{p_{ik}}{\sum_{l=1}^{K} p_{il}}$

    $q_{ik} \leftarrow \frac{p_{ik}^{1+\alpha}}{\left(\sum_{i \in \mathcal{Q}} p_{ik}^{1+\alpha}\right)^{1/2}} , i \in \mathcal{Q}$

    $q_{ik} \leftarrow \frac{q_{ik}}{\sum_{l=1}^{K} q_{il}}$

    $\boldsymbol{w}_k \leftarrow \dfrac{\dfrac{\lambda}{1+\alpha} \sum_{i \in \mathcal{S}} (y_{ik} \boldsymbol{z}_i + p_{ik}(\boldsymbol{w}_k - \boldsymbol{z}_i)) + \dfrac{|\mathcal{S}|}{|\mathcal{Q}|} \sum_{i \in \mathcal{Q}} (q_{ik} \boldsymbol{z}_i + p_{ik}(\boldsymbol{w}_k - \boldsymbol{z}_i))}{\dfrac{\lambda}{1+\alpha} \sum_{i \in \mathcal{S}} y_{ik} + \dfrac{|\mathcal{S}|}{|\mathcal{Q}|} \sum_{i \in \mathcal{Q}} q_{ik}}$

**end**

**Result:** Query predictions $\hat{y}_i = \arg\max_k p_{ik} , i \in \mathcal{Q}$

---

**Algorithm 2:** TIM-GD

**Input** : Pre-trained encoder $f_\phi$, Task $\{\mathcal{S}, \mathcal{Q}\}$, # iterations $iter$, Temperature $\tau$, Weights $\{\lambda, \alpha\}$, Learning rate $\gamma$

$\boldsymbol{z}_i \leftarrow \frac{f_\phi(\boldsymbol{x}_i)}{\|f_\phi(\boldsymbol{x}_i)\|_2} , i \in \mathcal{S} \cup \mathcal{Q}$

$\boldsymbol{w}_k \leftarrow \frac{\sum_{i \in \mathcal{S}} y_{ik} \boldsymbol{z}_i}{\sum_{i \in \mathcal{S}} y_{ik}} , k \in \{1, \ldots, K\}$

**for** $i \leftarrow 0$ **to** $iter$ **do**

    $p_{ik} \leftarrow \exp\left(-\frac{\tau}{2} \|\boldsymbol{w}_k - \boldsymbol{z}_i\|^2\right)$

    $p_{ik} \leftarrow \frac{p_{ik}}{\sum_{l=1}^{K} p_{il}}$

    $\boldsymbol{w}_k \leftarrow \boldsymbol{w}_k - \gamma \nabla_{\boldsymbol{w}_k} \mathcal{L}_{\text{TIM}}$

**end**

**Result:** Query predictions $\hat{y}_i = \arg\max_k p_{ik} , i \in \mathcal{Q}$

---

## C  Summary figure

We hereby provide a summarizing figure of the training and inference stages used in TIM.

Figure 2: Outline of TIM framework (best viewed in color). First, the feature extractor is trained with the standard cross-entropy on the base classes. Then, it is kept fixed at inference and weights $\mathbf{W}$ are optimized for by minimizing the cross-entropy on the support set $\mathcal{S}$, while maximizing the mutual information between features and predictions on the query set $\mathcal{Q}$.

## D  Details of ADM ablation

In Table 6, we provide the $\mathbf{W}$ and $\boldsymbol{q}$ updates for each configuration of the TIM-ADM ablation study, whose results were presented in Table 4. The proof for each of these updates is very similar to the proof of Proposition 2 detailed in Appendix A. Therefore, we do not detail it here.

Table 6: The $\mathbf{W}$ and $\boldsymbol{q}$-updates for each case of the ablation study. "-" refers to the updates in Proposition 2. "NA" refers to non-applicable.

| Loss | $w_k$ **update** | $q_{ik}$ **update** |
|------|------------------|---------------------|
| CE | $\dfrac{\sum\limits_{i \in \mathcal{S}} y_{ik} \boldsymbol{z}_i}{\sum\limits_{i \in \mathcal{S}} y_{ik}}$ | N/A |
| $\mathrm{CE} + \widehat{\mathcal{H}}(Y_\mathcal{Q}|X_\mathcal{Q})$ | - | $\propto p_{ik}^{1+\alpha}$ |
| $\mathrm{CE} - \widehat{\mathcal{H}}(Y_\mathcal{Q})$ | - | $\propto \dfrac{p_{ik}}{\left(\sum\limits_{i \in \mathcal{Q}} p_{ik}\right)^{1/2}}$ |
| $\mathrm{CE} - \widehat{\mathcal{H}}(Y_\mathcal{Q}) + \widehat{\mathcal{H}}(Y_\mathcal{Q}|X_\mathcal{Q})$ | - | - |

## Footnotes

[8]Note that strong duality holds since the objective is convex and the simplex constraints are affine. This means that the solutions of the (KKT) conditions minimize the objective.