[Reviews · NeurIPS 2020]

Review 1

Summary and Contributions: This paper presents a transductive few-shot learning method with information maximization. The aim is to maximize the mutual information between the query and the support set for each episode of a few-shot learning task. The idea is to use features from the pre-trained model and update the weights of classifiers using the query and the support images. The performance is superior compared to the recent transductive few-shot learning methods such as TPN and SIB.

Strengths: - The proposed is somewhat novel for few-shot learning taking the perspective from mutual information. - The results are superior compared to the previous methods.

Weaknesses: - The motivation of the method is weak. Especially in introduction, the reference to the mutual information is InfoMax in deep representation learning. While the paper uses a fully-supervised model (pretrained) and the proposed method has less correlation to the input and output of deep networks presented in DeepInfoMax. - Some citations are missing for the recent works relating to transductive few-shot learning and information maximization. It would be recommended to discuss [1] and [2]. - The method shares similarity to [3], especially Eq. 7 in [3]. Would the authors highlight and discuss the difference? - In the 1-shot case, is the one-hot encoded label directly used for $\pi$ in Eq. 3? - It is not described well how w is updated. In TIM-GD, is w updated always with all of the classes (using the loss in Eq. 3)? Because in some recent works for few-shot learning, the samples to update w are sampled (like mini-batch) from the support set and not necessarily containing all samples from all classes (K). - Is this proposed method limited only to the uniform distribution? It is still not very clear how this method can be applied to the iNat dataset with highly-imbalanced tasks. [1] Li et al., "Learning to Self-Train for Semi-Supervised Few-Shot Classification," Neurips, 2019. [2] Guo and Cheng, "Attentive Weights Generation for Few Shot Learning via Information Maximization," CVPR, 2020. [3] Hu et al., "Empirical Bayes Transductive Meta-Learning with Synthetic Gradients," ICLR, 2020.

Correctness: The equations are correct and the experiments also follow the existing works.

Clarity: The rebuttal clarifies the ambiguous segments. Please reflect the unclear statements in the revised version.

Relation to Prior Work: Please discuss [2, 3] in the revised version as the readers might need a better description to bridge the existing techniques and the proposed approach.

Reproducibility: Yes

Additional Feedback: After going to the rebuttal and other reviews, I believe this work has benefits to the community. The rebuttal addressed my concerns, thus I vote for acceptance. The other reviewers also ask for detailed explanation about the proposed approach and experiments. I encourage the authors to revise the manuscript based on the concerns from the other reviewers also and make it clear.


Review 2

Summary and Contributions: The authors propose a kind of information maximization method for few-shot learning. Different from traditional methods, the authors integrate a KL divergence regularizer to maximize the common information between query samples and predictions. According to the experiments, this method can deal with few-shot tasks, where there even exists domain-shift.

Strengths: 1. This paper proposes a method inspired by information theory, which is nicely theoretical guaranteed. 2. The authors derive an approximate method to minimize the whole loss function, which can effectively speed up the optimization. 3. Extensive experiments have been conducted.

Weaknesses: 1. When using the standard few-shot setting (e.g., 5-way 5-shot tasks), \pi is a uniform distribution for both support set and query set under a transductive setting. In this case, there is no difference between \pi and \pi_k (fixed by the label-statistic information from the support labels). In other words, \pi_k is just a uniform distribution. Eq(2) degrades to the standard mutual information as mentioned in Remark 1. In fact, \pi is an estimation of the real distribution of the query set, which is not known in the real setting, i.e., an inductive setting. Importantly, I don’t think \pi can be estimated by the prior distribution of the support set. They may be different from each other. The authors estimate the distribution of the query set by using the distribution of the support set under a transductive setting. It’s not reasonable. I think it is just an artificial coincidence. 2. In Eq (1), the authors say p_ik is proportional to exp(*). Why? Does the feature vector have been \ell_2 normalized? 3. I think it needs some figures or some words to introduce the process of the algorithm. 4. The implementation details are not very clear. For example, what’s the image size used in the experiments.

Correctness: Maybe not.

Clarity: I think the writing needs to be improved. It is difficult for me to understand the whole process of the proposed method. The problem definition is not very clear. There are no figures or materials to help understand.

Relation to Prior Work: Yes, as I know, I think this paper differs from prior works.

Reproducibility: Yes

Additional Feedback: I have carefully read the authors' feedback and comments of other reviewers. Some of my concerns have been well addressed, but others have not. However, because this paper indeed have some impacts to the community, I would like to accept this paper and encourage the authors to refine the manuscript according to the following concerns. 1. What's the main contribution? In the feedback, the authors said "the use of a label marginal prior (R2, R4), which is just a convenient generalization of our MI, but not really the main contribution of the paper" (Lines 5-6) and "We emphasize that the MI (no prior) alone is our contribution, and achieves SOTA results over 4 standard benchmarks by wide margins (it is also competitive on iNat)" (Lines 14-15). However, in the submitted paper, the authors said "In fact, if we remove the marginal divergence D_KL in objective (3), our TIM objective reduces to the loss in [6]." (Lines 129-130 in the submitted paper). If my understanding is correct, MI (no prior) means there is no the D_KL term. Therefore, what's the contribution of this paper compared to [6]? 2. The introduced D_KL is really important? The authors said "The label-marginal regularizer D_KL is of high importance. As it will be observed from our experiments, it brings substantial improvements in performances (e.g., up to 10% increase in accuracy over entropy fine-tuning on the standard few-shot benchmarks)" (Lines 131-132 in the submitted paper). However, from the ablation study in Table 3, we can see that TIM-ADM (CE+D_KL+H) have almost the same results as TIM-ADM (CE+H) on the 5-shot setting on all three datasets. The slightly improvements on the 1-shot setting may come from the randomness. Also, the same phenomenon can be seen in Table 4, TIM-GD (Uniform) has almost the same results as SimpleShot [42]. 3. Why using a support-based prior? Although the authors have explained the reason of using support-based prior, it still cannot convince me because it's under a transductive setting. 4. Implementation details I cannot agree with the viewpoint of the authors. I think a good paper should be self-contained. Why not make this part clearer in your own paper?


Review 3

Summary and Contributions: This paper studies the transductive setting of few-shot learning from the mutual information maximization perspective. In particular, this paper designs TIM loss which consists of CE loss and empirical prior-aware mutual information. To train the model, this paper proposes TIM-GD which has the best performance and further proposes TIM-ADM to speed up the training while maintain the accuracy. The experiments on standard few-shot learning and the scenarios with domain shifts and severe class imbalance show significant performance improvements.

Strengths: - This paper introduces empirical prior-aware mutual information which include the standard mutual information as a special case when the prior is the uniform distribution. - The training procedure tries to maximize the empirical prior-aware mutual information and minimize the cross entropy simultaneously. The former is to use the supervision information from the support set, while the latter is to use the unsupervision information from the query set. - For the optimization, to speed up the training procedure, this paper designs Alternating direction method for minimizing the loss function, which achieves almost equal accuracy to the gradient descent.

Weaknesses: - This paper focuses on the transductive few-shot learning. However, such setting is too artificial, since the model can see all test data (not only one test task) during the training. - The loss function of this paper includes three terms: standard CE loss, KL divergence between prior and latent label distribution, and conditional entropy which makes classifier's coundaries not go through the dense regions of samples. The novelty is on the borderline.

Correctness: Yes. I haven't seen anything wrong so far.

Clarity: Yes.

Relation to Prior Work: Yes.

Reproducibility: Yes

Additional Feedback:


Review 4

Summary and Contributions: This paper presents a novel few-shot learning framework using transduction information Maximization. In the transduction setting, the authors assume the model can see the unlabeled testing data. They also introduce two new loss functions, marginal divergence and conditional entropy. The results show this model outperform the state-of-the-art in a large margin.

Strengths: 1. This model can significantly outperform the baselines on four different datasets 2. The paper shows the naive gradient descent optimization is two order of magnitude slower than baseline. To speed up the model, they introduce the alternating direction method using auxiliary variables to help training.

Weaknesses: The results in this paper are really promising. And the idea is novel. I just have some questions and suggestions about this paper: 1. In table 3, have the authors done multiple runs to get average accuracy? I just found TIM-GD+CE+{w} results are different from TIM-ADM+CE+{w}. I think they are supposed to be the same. 2. The entropy loss is well used on domain adaption, but I haven't found papers using the label prior and mutual information. Is it possible to apply this method to the domain adaptation problem? 3. In the paper, the authors assume the prior is uniform or S-based. However, in some cases, the query images might not satisfy this prior. For example, 50% of support images are class A, but only 10% of support images are class A. How will the model work on it? 4. Missing figure 5.

Correctness: Yes

Clarity: The paper is clear, but there are some typos. 1. In the abstract, "Transductive Infomation Maximization" should be "transduction information Maximization". 2. Line 276, "similar as " should be "similar to".

Relation to Prior Work: Yes

Reproducibility: Yes

Additional Feedback: The authors have addressed my questions. I will keep my score unchanged.

[Author Response · NeurIPS 2020]

**AC and all Reviewers**: We thank all reviewers. To summarize, all the reviewers acknowledged the novelty of our
transductive Mutual Information (MI) loss, the novelty of the ADM optimizer and the speed-up it brings, as well as the
SOTA results over 5 benchmarks, by significant margins. The criticisms are essentially based on: discussions on recent
prior works and motivation (R1), a misunderstanding of the standard transductive setting in few-shot learning (R3), and
the use of a label marginal prior (R2, R4), which is just a convenient generalization of our MI, but not really the main
contribution of the paper (our prior-free MI loss obtains SOTA results over all benchmarks).

**(R2-R4) Concerns about the use of a prior $\pi$ over label marginals Y**: First, we concede that writing Eq. (3) as we
did may have conveyed that the prior itself is crucial to the well functioning of the method, and needs to be estimated
accurately from the support set. We discuss the motivation of using $\mathcal{S}$-based prior later. Regardless, this does not
alter the main contribution of the work, which is the MI (uniform prior). As a matter of fact, R4 also brought up this
point, yet gave an accept score (7). The main utility of the term $\mathcal{D}_{\text{KL}}$ is actually to prevent the trivial solutions of
conditional-entropy minimization, as discussed in L.229-232, rather than to impose exact label proportions. Therefore,
the exactitude of the prior is not crucial. In fact, using any prior at all (i.e., non-uniform) is optional in our formulation,
as it merely represents a generalization of the standard MI. We emphasize that the MI (no prior) alone is our contribution,
and achieves SOTA results over 4 standard benchmarks by wide margins (it is also competitive on iNat). Imposing
a prior when available (could come from any source) is application-dependent and can indeed lead to enhanced
performances, but is, again, not necessary. All in all, the issue could be easily fixed by writing CE + MI as our main
objective (3), and merely proposing the prior-aware MI version as an extension when a prior is available. **Why using**
**a support-based prior?** The very setting of FSL assumes that $\mathcal{S}$ and $\mathcal{Q}$ are sampled from the same underlying joint
distribution $p(x, y)$ (Section 1.2 in [Wang et al. "Generalizing from a few examples: A survey on few-shot learning."
ACM Computing Surveys]), which uniquely determines the marginal $p(y)$ (as can be seen by simply marginalizing
out x). Hence, empirical marginals should be close $\hat{p_{\mathcal{S}}}(y) \approx \hat{p_{\mathcal{Q}}}(y)$. Therefore, we concede to R2-R4 that exactly
having $\hat{p_{\mathcal{S}}}(y) = \hat{p_{\mathcal{Q}}}(y)$ as in standard benchmarks may appear somewhat artificial. Typically, the $\mathcal{S}$-based prior does
not exactly match the true query marginal as in the more *realistic* iNat task, but still provides a better estimate of it (as
shown by our results on iNat).

**(R1) Clarification of the motivation** : We are maximizing the MI between the query features and predictions (i.e.,
between inputs and outputs), not between the query and support sets. We mentioned DeepInfoMax as an example of a
deep-learning instance of the original InfoMax principle. We use the latter principle in a different way: The inputs
are considered as extracted features of pretrained network, and output are predictions. Such idea is motivated by and
relates to the MI in classical clustering works (e.g., [16]), which we cite/discuss in L.62-64. **Significant differences**
**with references [1, 2, 3]**: We start by emphasizing that [1], [2] and [3] design meta-learning methods for both training
and inference. Our method is used for inference only, and can work on top of any pre-trained feature extractor. As for
the objective functions, Ref. [1] doesn't deal with the MI. The self-labeling in [1] can be viewed as a minimization
of the min-entropy of query predictions (we will cite/discuss this work). Also, the MI measures in [2, 3] evaluate
quantities that are completely different from us because [2,3] learn conditional distributions over classifier's weights,
while we learn task-specific weights by direct optimization. Specifically, [2] maximizes the MI between features and
weights while we maximize MI between the query features and labels. The former is intractable as both variables
are continuous with no access to the underlying distributions, and requires a variational approximation, while the
latter (ours) is tractable ($Y_{\mathcal{Q}}$ is low-dimensional/discrete + we have access to $p(y|f_\phi(x))$). Objective (7) in [3] uses a
cross-entropy (CE) on query samples while our objective uses CE on support and an unsupervised MI on query samples.
As for the KL term in (7), it encourages the posterior over weights to match a prior $p_\psi$, while the KL in our objective
(3) encourages the label marginals to be close to a prior. Notice that all 3 papers require additional modules to train
(soft-weighting network in [1], reconstruction in [2] and gradient synthetizer in [3]), while our method doesn't (fewer
parameters/hyper-parameters). **W-updates:** $W$ are updated with all samples (both support and query) at once (possible
because we work directly on low-dimensional extracted features). **Is the method limited to uniform distribution?**
No, we can use any prior in Eq. (2). While $\pi$ is uniform for all the standard benchmarks, for iNat, we showed that a $\pi$
estimated from the labeled support samples yields improvements (Table 4). **1-shot case, is the one-hot encoded label**
**used for $\pi$ in Eq. (3)**: No. In all our 1-shot experiments, $\pi$ is uniform (i.e., no prior – we minimize the MI).

**(R3) Mis-understanding of the transductive few-shot setting:** The model **does not see test data** during training
(only base classes are seen during training). At test-time, it only sees **one test task at a time** . As discussed in lines
37-56, this transductive setting is now a standard in few-shot learning, as evidenced by the large number of recent
major-conference publications in this setting, e.g., [6, 13, 14, 18, 23, 30], among many others.

**(R2) (Implementation details:)** As mentioned in lines 196-200, we used the standard training in [42] for the base
classes. We also used standard image data sets. This is why we did not provide those training and image-size details.

**(R4) ADM vs. GD**: All results are averaged over 10'000 runs. For the CE row in Table 3, results differ because TIM-GD
performs GD based finetuning, while TIM-ADM uses the updates in Table 9 (appendix). **Application to domain**
**adaptation:** Yes, applying our method to domain adaption might be very interesting. Thanks for the suggestion!

[Meta-Review · NeurIPS 2020]

The reviewers unanimously agree that the introduction of mutual information into the few-shot setting is a promising direction and of interest to the few-shot learning community. There are still some concerns about the effectiveness of the KL term as pointed out by R2 that should be addressed, and it would be good to make the paper as self-contained as possible, as suggested by R1.